# A Genome-Wide Association Study of Biomass Yield and Feed Quality in Buffel Grass (*Cenchrus ciliaris* L.)

**Alemayehu Teressa Negawo [1], Meki Shehabu Muktar [1], Ricardo Alonso Sánchez Gutiérrez [1,2], Ermias Habte [1], Alice Muchugi [1] and Chris S. Jones [1,3,*]**

1   Feed and Forage Development Program, International Livestock Research Institute (ILRI),
    Addis Ababa P.O. Box 5689, Ethiopia; a.teressa@cgiar.org (A.T.N.); m.shehabu@cgiar.org (M.S.M.);
    rasanchez.gutierrez@gmail.com (R.A.S.G.); e.habte@cgiar.org (E.H.); a.muchugi@cgiar.org (A.M.)
2   Instítuto Nacional de Investigaciones Forestales Agrícolas y Pecuarias, Campo Experimental Zacatecas,
    Calera 98500, Zacatecas, Mexico
3   International Livestock Research Institute (ILRI), P.O. Box 30709, Nairobi 001000, Kenya
*   Correspondence: c.s.jones@cgiar.org

**Abstract:** The development of modern genomic tools has helped accelerate genetic gains in the breeding program of food crops. More recently, genomic resources have been developed for tropical forages, providing key resources for developing new climate-resilient high-yielding forage varieties. In this study, we present a genome-wide association study for biomass yield and feed quality traits in buffel grass (*Cenchrus ciliaris* L. aka *Pennisetum ciliare* L.). Genome-wide markers, generated using the DArTSeq platform and mapped onto the *Setaria italica* reference genome, were used for the genome-wide association study. The results revealed several markers associated with biomass yield and feed quality traits. A total of 78 marker–trait associations were identified with $R^2$ values ranging from 0.138 to 0.236. The marker–trait associations were distributed across different chromosomes. Of these associations, the most marker–trait associations (23) were observed on Chr9, followed by Chr5 with 12. The fewest number of marker–trait associations were observed on Chr4 with 2. In terms of traits, 17 markers were associated with biomass yield, 24 with crude protein, 26 with TDN, 14 with ADF, 10 with NDF and 6 with DMI. A total of 20 of the identified markers were associated with at least two traits. The identified marker–trait associations provide a useful genomic resource for the future improvement and breeding of buffel grass.

**Keywords:** climate change; marker-assisted breeding; tropical forage; forage improvement; genetic resources; drought tolerance

## 1. Introduction

Achieving improved livelihoods, reduced poverty and reduced malnutrition in the world would be very difficult without addressing the challenges of sustainable livestock production in low- and middle-income countries (LMIC). Livestock play multiple crucial roles in rural livelihoods and the economy of LMIC [1–3] where smallholder farmers account for most of the crop–livestock production. Under smallholder farmers' conditions, natural pasture is the main source of feed for livestock animals and, among others, feed resources are the major limiting factor for livestock production and productivity [4]. Hence, there is a strong need to increase feed resource availability through the development of climate-resilient, low-input forage varieties that provide better yields of quality forage in the current trend of climate change and enable expanding livestock production to marginally suitable areas and agroecological conditions.

Among agricultural technologies, plant breeding has played a considerable role in crop yield improvements over the last several decades [5]. In the past few years, the development and integration of modern genomic tools has benefited plant breeding programs [6] and contributed to the development of new varieties of major food crops. In more recent

years, genomic resources have also been developed for a limited number of important species of tropical forages. For example, during the last few years, at the International Livestock Research Institute, genome-wide markers were generated for Napier grass, buffel grass, Rhodes grass, lablab and *Sesbania sesban* [7–10] and are being developed for *Urochlao* spp. and *Megathyrus maximus* (unpublished data). Similarly, high-throughput genome-wide markers have also been developed for tropical forages elsewhere [11–13]. Reference genomes have also been developed for a few of the key tropical forage crops [14–20]. These genomic resources have been used for the analysis of genetic diversity, subsetting, genome-wide association and population genetic studies, and will continue to be useful tools and resources for tropical forage research and development. The integration of these genomic tools into field screening and evaluation will enable efficient and accelerated forage breeding programs to develop adaptive and climate-resilient varieties to transform livestock production in tropical regions.

Among the tropical forages, buffel grass is an important drought-tolerant perennial species [21] grown throughout the tropical and subtropical regions of the world [21,22]. It is an apomictic species with a basic chromosome number of nine and three ploidy levels: tetraploid (2n = 4x = 36), pentaploid (2n = 5x = 45) and hexaploid (2n = 6x = 54) [23,24]. It is an important grass cultivated as a pasture or for hay production [25]. Under no input production conditions, it can produce up to 18 t DM (dry matter)/ha/annum [25] and forage with a crude protein content of 6–16% [22]. Buffel grass has been reported to produce DM yields of up to 12 t/ha in Kenya [26,27], 8 t/ha in USA [28], 7 t/ha in Pakistan [29] and 21 t/ha in Ethiopia [30]. Improved forage varieties that are better adapted to produce more quality biomass across a range of agroecologies and production systems are a prerequisite and are required more than ever for supporting enhanced livestock production in a sustainable manner [31]. Despite limited improvement efforts, conventional tropical forage breeding programs have contributed to the development of improved forage cultivars with a number of buffel grass cultivars developed over the last few decades [22]. However, genetic gains from conventional tropical forage breeding programs have been low, particularly in view of the growing demand for animal source foods globally, and breeding programs should leverage the combination of phenotyping, genotyping, and envirotyping strategies in order to increase genetic gains and help secure the future of livestock production in the tropics [31]. The International Livestock Research Institute (ILRI) holds a large collection of buffel grass germplasm collected from different countries in Africa and Asia [9]. Agro-morphological studies have revealed the diversity embedded in the buffel grass collection maintained in the forage genebank at ILRI [32,33]. Facultative apomictic lines that could offer a potential resource for forage breeding programs to generate new and improved varieties have also been identified in the collection [34]. Recently, we generated a large set of genome-wide markers using a next generation sequencing approach and reported on the large amount of genetic diversity held in the collection [9]. To our knowledge, there have been no genomic studies in buffel grass that combine phenotypic and genotypic data analysis to investigate the crops' genetic architecture. Thus, in this current study, we envisage filling this gap by employing a genome-wide association study for biomass yield and feed quality traits. Here we leveraged the data generated from previous agro-morphological [33] and genotyping studies [9], and present a genome-wide association study (GWAS) for biomass yield and feed quality traits in buffel grass.

## 2. Materials and Methods

### 2.1. High-Density Genome-Wide Markers

Genome-wide SNP and SilicoDArT markers, generated using the DArTSeq platform [35], were used in this study. The markers were mapped onto the *Setaria italica* reference genome [36] as described previously [9].

## 2.2. Phenotypic Data

The phenotypic data were collected from a field experiment as described elsewhere [33] and used for GWAS. Briefly, the experiment was established in 2014 during the main rainy season at the Bishoftu field genebank, Bishoftu, Ethiopia [33]. All the accessions evaluated in the experiment were obtained from the Zwai field genebank, Zwai, Ethiopia. The passport data of the accessions are provided in Supplementary Table S1. The experiment was conducted using a randomised complete block design with three replications, and data were collected during the main growing seasons (August–September) of 2015 and 2016. During both growing seasons, the plants were clean cut at 10 cm above ground (August 18th), and the data were collected 45 days after the clean cut. The data collected include biomass yield (YLD, kg/ha/annum), plant height (PH, cm), crude protein (CP, % of dry matter (DM)), Neutral Detergent Fibre (NDF, % of DM), Acid Detergent Fibre (ADF, % of DM), total digestible nutrients (TDN, % of DM) and dry matter intake (DMI, % of body weight). For biomass yield estimation, the plants within a quadrat with an area of one square metre were harvested at 10 cm above the ground and weighed immediately, and the weight was converted to yield per hectare. For plant height, three plants per plot were measured from the ground to the tip of the tallest inflorescence, and the average was used for further analysis. For feed quality analysis, 300 g of freshly harvested material was oven dried (72 h at 55 °C), ground to pass through a one mm sieve and used for Near Infrared Spectroscopy (FOSS Forage Analyzer 5000 with software package WinISI II) (NIRS) scanning, as described previously [37]. TDN and DMI were estimated using ADF and NDF values from the NIRS data using the equations TDN = 88.9 − (0.779 × ADF) and DMI = 1.2/NDF × 100 [38].

## 2.3. Data Analysis

A normality test analysis based on the Shapiro-Wilk method was conducted using the R package nortest (version 1.0.4) [39]. Statistical analysis was conducted using analysis of variance (ANOVA) in R software (version 4.2.2) to determine the significance of the main effects and the interactions using the following model:

$$Y_{ijk} = \mu + G_i + B_j + T_k + (G_i \times T_{ik}) + \varepsilon_{ijk} \tag{1}$$

where $Y_{ijk}$ is the response, $\mu$ = overall mean, $G_i$ = effect of the *i*th buffel grass genotype, $B_j$ = effect of the *j*th Block effect, $T_k$ = effect of the *k*th growing season, $G \times T_{ij}$ = the interaction of *i*th genotype and *j*th growing season and $\varepsilon_{ijk}$ = the residual error. The least significant difference (LSD) test for a comparison of the mean values of traits was employed to compare genotypes for traits with significant differences. Genetic parameters, genotypic coefficient of variation (*GCV*) and phenotypic coefficient of variation (*PCV*) were estimated using the formulae [40]

$$GCV = \frac{\sqrt{\sigma g^2}}{X} \times 100 \tag{2}$$

$$PCV = \frac{\sqrt{\sigma p^2}}{X} \times 100 \tag{3}$$

where *GCV* = genotypic coefficient of variation, *PCV* = phenotypic coefficient of variation, $\sigma g^2$ = genotypic variance, $\sigma p^2$ = phenotypic variance and *X* = grand mean.

## 2.4. Marker–Trait Association Analysis

A Bartlett test, using the bartlett.test() function of the R package Stats (version 4.3.2) [41], was used to assess the homogeneity of error variance prior to pooling the data for the GWAS. The GWAS were performed as described by Muktar et al. [42] using fixed and random model Circulating Probability Unification (FarmCPU) [43], Bayesian information and Linkage-disequilibrium Iteratively Nested Keyway (BLINK) [44] and General linear model (GLM) algorithms [45], implemented in the R package Genomic Association and Prediction integrated tool version 3(GAPIT3) [46]. Linkage maps of the markers associated with traits of interest were generated using the R package LinkageMapView (version 2.1.2) [47].

To assess the putative functional genes underlying the genomic regions of the identified marker–trait associations, an NCBI blast search was conducted using the sequence of the markers.

## 3. Results

Genotyping data were available for 205 accessions [9], whereas phenotypic evaluation data were available for 126 accessions. When these resources were combined, 120 accessions had both genotypic and phenotypic data, and, hence, data from these 120 accessions were considered for the marker–trait association studies. The normality test analysis showed that the agronomic and feed nutrition trait data were normally distributed. Following a normality test, outliers were removed, resulting in 110 and 114 accessions being used for the GWAS for the 2015 and 2016 growing seasons, respectively. The homogeneity of variance test showed that there was a significant difference between the 2015 and 2016 season data for all traits except for CP and NDF; hence, the GWAS analysis was conducted for the 2015 and 2016 seasons separately, as well as after combining the two growing seasons' data.

### 3.1. Variation in Biomass Yield, Plant Height and Feed Quality Traits of Buffel Grass Accessions

The average biomass yield per annum in 2015 and 2016 was 3231.47 and 5926.96 kg/ha, respectively, whereas the two seasons combined mean biomass yield per annum was 4562.22 kg/ha. Figure 1 shows the boxplot visualisation of the distribution and outliers of the data for biomass yield, plant height and feed quality traits by growing seasons. The mean performance of each accession over the two growing seasons for agronomic and feed quality traits are presented in the Supplementary information (Supplementary Tables S2 and S3). The mean plant height was 85.58 cm, 121.18 cm and 103.31 cm for 2015, 2016 and the combined seasons, respectively. The mean value for crude protein (CP) was 12.49%, 8.33% and 10.02% for 2015, 2016 and the combined data, respectively. The mean value for NDF and ADF for the combined seasons was 72.37% (70.42% for 2015 and 73.78% for 2016) and 43.77% (40.31% for 2015 and 47.78% for 2016), respectively. Similarly, the mean value for TDN and DMI over the two seasons was 48.03% (52.54% for 2015 and 44.72% for 2016) and 1.66% (1.71% for 2015 and 1.63% for 2016), respectively.

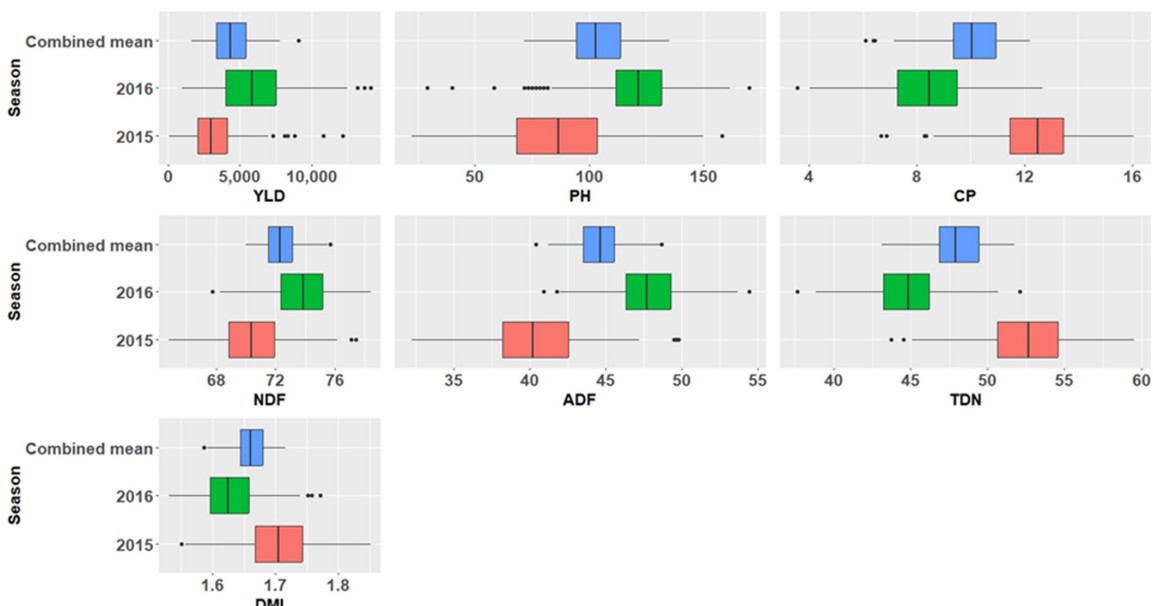

**Figure 1.** Boxplot visualisation showing the distribution and outliers of the data for biomass yield (YLD, kg/ha/annum), plant height (PH, cm) and feed quality traits by growing seasons. The red, green and blue boxes are for seasons 2015, 2016 and the combined mean, respectively. CP = Crude protein, NDF = Neutral Detergent Fibre, ADF = Acid Detergent Fibre, TDN = total digestible nutrients and DMI = dry matter intake.

### 3.2. Effect of Genotype and Seasonality on Buffel Grass Forage Performance

ANOVA results for all of the traits revealed highly significant (<0.001) differences among genotypes, blocks and seasons. The genotype–season interaction was not significant except for NDF (Table 1). Overall, the results showed that the performance of buffel grass was primarily affected by the genotype and season of production. The significant difference for the block effect shows that blocking was effective in reducing the soil heterogeneity.

**Table 1.** ANOVA summary for agronomic and feed quality traits from 126 buffel grass accessions in 2015 and 2016 growing seasons at Bishoftu, Ethiopia.

| Traits/Sources of Variation | YLD | PH | NDF | ADF | CP | TND | DMI |
|---|---|---|---|---|---|---|---|
| Genotype | <0.001 | <0.001 | <0.001 | <0.001 | <0.001 | <0.001 | <0.001 |
| Replication | <0.001 | <0.001 | <0.001 | <0.001 | <0.001 | <0.001 | <0.001 |
| Season | <0.001 | <0.001 | <0.001 | <0.001 | <0.001 | <0.001 | <0.001 |
| Genotype: Season | NS | NS | NS | 0.001 | NS | NS | NS |
| CV% | 34.9 | 17.9 | 2.3 | 4.7 | 13.4 | 4.1 | 2.3 |
| R-square % | 73 | 73 | 77 | 88 | 85 | 89 | 78 |

Coefficient of variation (CV), YLD = Biomass yield, PH = Plant height, CP = Crude protein, NDF = Neutral Detergent Fibre, ADF = Acid Detergent Fibre, TDN = total digestible nutrients, DMI = dry matter intake and NS = non-significant.

### 3.3. Correlation of Phenotypic and Feed Quality Traits

Figure 2 shows correlation coefficients between yield, plant height and nutritional quality traits among the buffel grass accessions for the two growing seasons and the combined data. There was a positive and significant correlation among biomass yield, plant height, NDF and ADF. Similarly, a positive and significant correlation was observed among CP, TDN and DMI. On the other hand, CP and DMI had a negative and significant correlation with biomass yield, plant height, NDF and ADF.

### 3.4. Quantitative Genetic Variation

The phenotypic coefficient of variation (PCV) and the genotypic coefficient of variation (GCV) were calculated to assess the contribution of the factors to the respective traits (Table 2). The PCV value for biomass yield was equivalent to the GCV values. PCV values for plant height and feed quality traits were higher than the corresponding GCV values.

**Table 2.** Variations and heritability for biomass yield, plant height and feed quality traits of 126 buffel grass accessions for 2015 and 2016 growing seasons at Bishoftu Ethiopia.

| Traits/Statistics | Minimum | Maximum | Mean | PCV | GCV |
|---|---|---|---|---|---|
| YLD (Kg/ha) | 1609.65 | 9097.54 | 4562.22 | 28.1 | 28.1 |
| PH (cm) | 71.50 | 135.22 | 103.31 | 13.9 | 9.9 |
| CP (%) | 6.11 | 12.21 | 10.02 | 32.8 | 8.9 |
| NDF (%) | 69.98 | 75.68 | 72.37 | 11.8 | 1.2 |
| ADF (%) | 40.40 | 48.69 | 44.62 | 15.2 | 2.8 |
| TND (%) | 43.10 | 51.73 | 48.03 | 14.7 | 2.9 |
| DMI (%) | 1.59 | 1.72 | 1.66 | 77.6 | 1.2 |

YLD = Biomass yield, PH = Plant height, CP = Crude protein, NDF = Neutral Detergent Fibre, ADF = Acid Detergent Fibre, TDN = total digestible nutrients, and DMI = dry matter intake, GCV = genotypic coefficient of variation and PCV = phenotypic coefficient of variation.

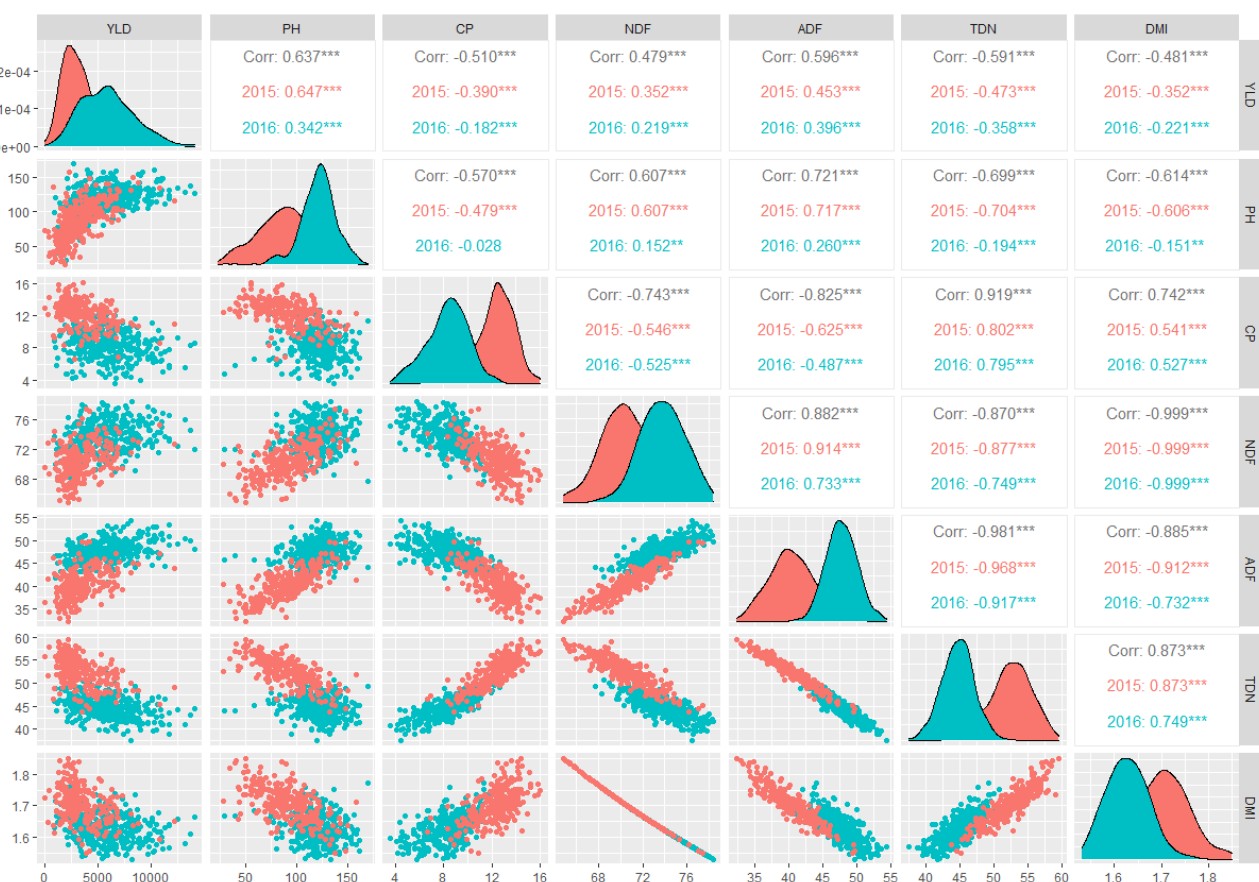

**Figure 2.** Correlation coefficient among agronomic and quality traits in the buffel grass collection. The correlation coefficient value for 2015 (red), 2016 (light blue) and combined (black) data are shown on the upper half part of the graph. YLD = Biomass yield, PH = Plant height, CP = Crude protein, NDF = Neutral Detergent Fibre, ADF = Acid Detergent Fibre, TDN = total digestible nutrients and DMI = dry matter intake. ** *p*-value < 0.01, *** *p*-value < 0.001.

### 3.5. Buffel Grass Accession Clustering Based on Phenotypic and Feed Quality Traits

Principal component analysis was used to group the buffel grass accessions based on phenotypic and feed quality trait data from individual growing seasons and the combined seasons. Figure 3 shows the clustering of buffel grass accessions based on phenotypic and feed quality traits from the two growing seasons. The first two principal components accounted for 87.2% of the total variation for the combined seasons' data. The PCA grouped the accessions into those with better biomass yield, better feed quality and poor feed quality accessions. For example, accessions such as 19,369, 13,406, 19,425, 19,467 and 12,884 were among the group of accessions with high crude protein content during both growing seasons. Accessions 19,459, 19,448 and 19,439 had the lowest CP content. Accessions 19,442, 6646 and 19,459 were among those with a high biomass yield, whereas accessions 15,688, 13,121 and 12,769 produced the lowest biomass yield. The tallest accessions were 13,461, 16,609 and 19,414, whereas the shortest accessions were 6645, 19,470 and 19,371. NDF contents were highest in accessions 13,461, 16,609 and 19,442, whereas accessions 6645, 19,420 and 19,367 contained the lowest NDF. Accessions 13,461, 19,462, 19,442, 19,448 and 16,609 were a few of those with high ADF (poor feed quality accessions), whereas accessions 12,769, 19,367 and 13,284 were among those with lowest ADF. The highest values for TDN and DMI was observed in accessions 12,769, 18,094 and 19,467, and lowest values were obtained from accessions 19,442, 13,461 and 19,448. Accessions 19,367, 12,769, 18,094, 19,425 and 19,420 were among the accessions with the highest DMI, whereas accessions 13,461, 16,609 and 19,442 were among those with the lowest DMI.

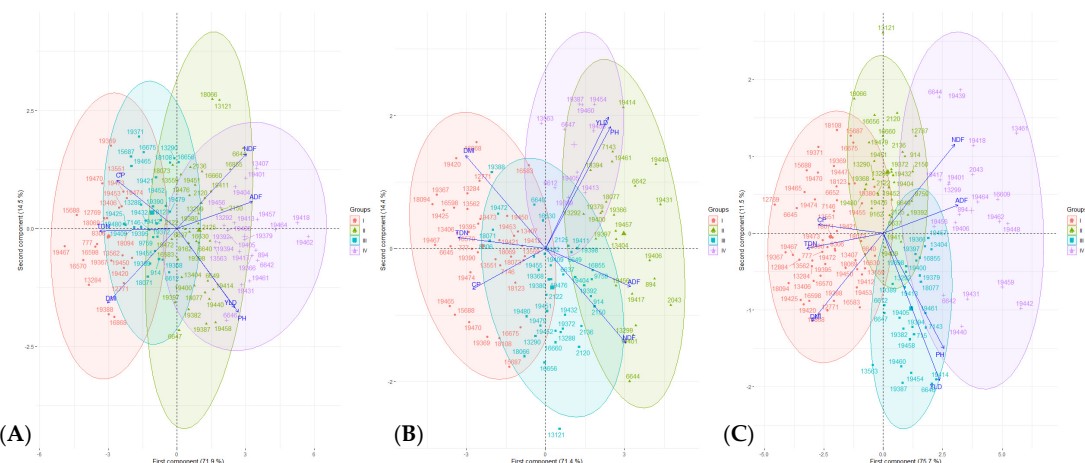

**Figure 3.** Clustering of accessions based on phenotypic and feed quality traits using (**A**) 2015, (**B**) 2016 and (**C**) combined growing seasons data. The first two principal components accounted for 86.4%, 85.8% and 87.2% of the total variation for 2015, 2016 and combined growing seasons, respectively. YLD: Biomass yield, PH: Plant height, NDF: Neutral Detergent Fibre, ADF: Acid detergent Fibre, CP: crude protein, TDN: Total digestible nutrients, DMI: Dry matter intake.

### 3.6. Performance of Genetic Clusters Identified Using DArTSeq Genome-Wide Markers

The performance of clusters identified using DArTSeq markers [9] were assessed. Figure 4 shows the performance of the different clusters (Supplementary Table S4). Cluster IV had the highest biomass yield followed by cluster VIII. Cluster II had the lowest biomass yield. Similarly, cluster IV had the tallest plants, whereas cluster II had the shortest plants. In terms of feed quality, clusters I, II and III were of a higher quality than the rest of the clusters. These three clusters had the highest CP (10.69–10.78%) and TDN (48.90–49.06%). Cluster IV had the lowest CP (9.95%) and TDN (46.82%). Cluster II had the highest TDN and DMI values.

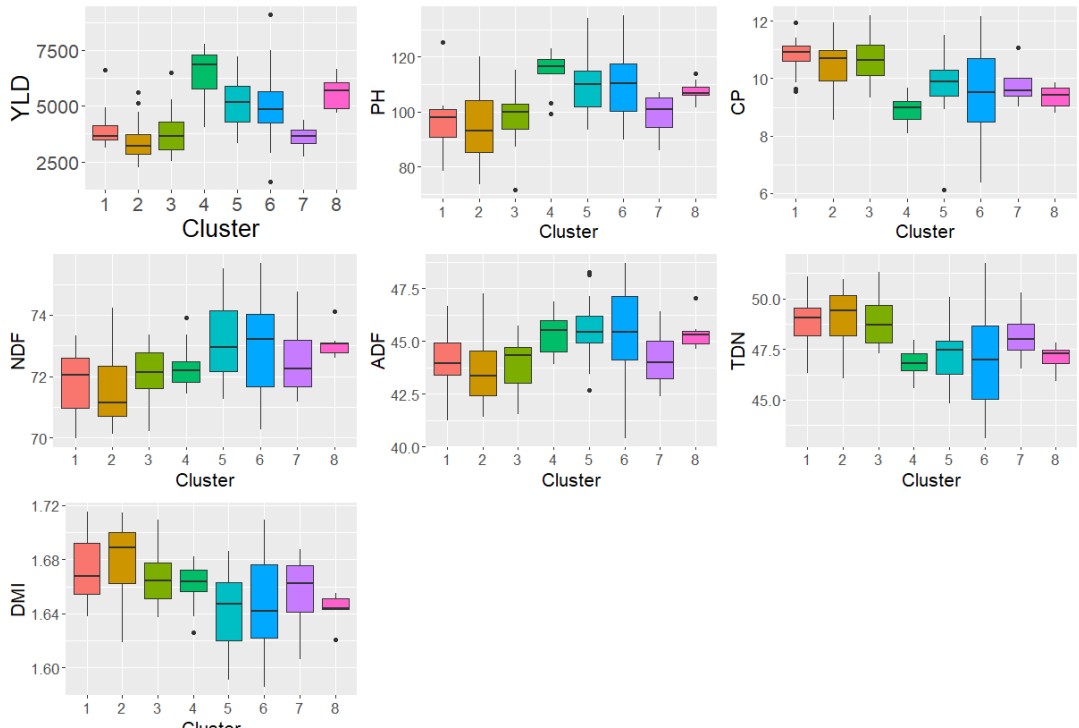

**Figure 4.** Performance of the clusters reported using DArTSeq markers [9]. The clusters are indicated on the *x*-axis, whereas the different traits are indicated on the *y*-axis. YLD = Biomass yield, PH = Plant

height, CP = Crude protein, NDF = Neutral Detergent Fibre, ADF = Acid Detergent Fibre, TDN = total digestible nutrients and DMI = Dry matter intake.

### 3.7. Genome-Wide Distribution and Density of Markers

The DArTSeq markers were mapped onto the *Setaria italica* reference genome [36]. Figure 5 shows the genome-wide distribution and density of the markers on the reference genome. These mapped markers were used for genome-wide association studies for the different traits. Accordingly, the total number of SNP and SilicoDArT markers used for GWAS was 7206 and 8342, respectively (Supplementary Table S5).

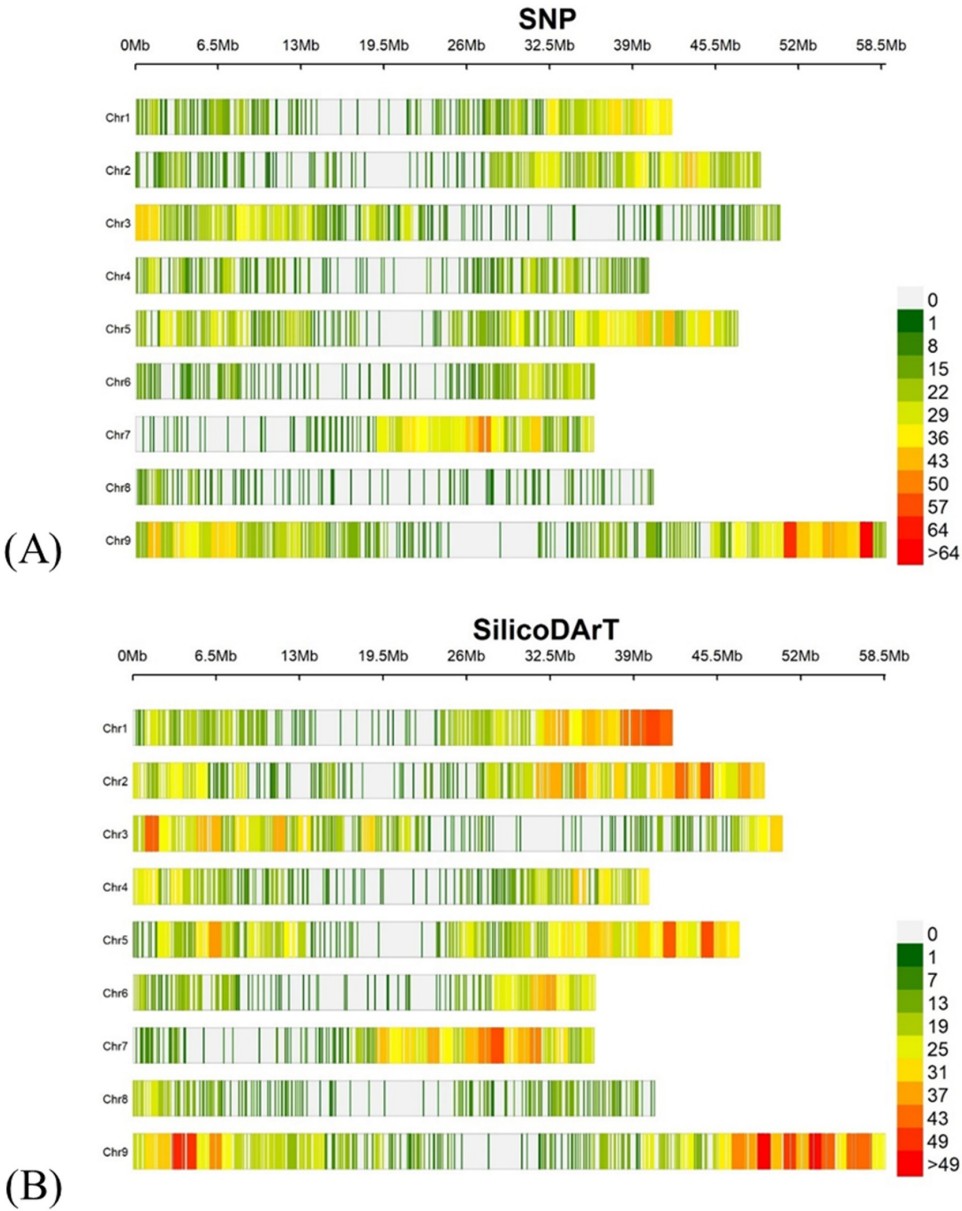

**Figure 5.** Chromosome-wide distribution and density of DArTSeq markers on the *Setaria italica* genome: (**A**) SNP markers and (**B**) SilicoDArT markers.

### 3.8. Data Filtering for Association Studies

For association studies, markers with known genomic positions were used. The markers were also filtered for missing data ($\leq$20%), polymorphic information content ($\geq$0.2) and minor allele frequency (MAF, 0.05). The phenotypic and feed quality data were

checked for normality distribution (Supplementary Figure S1), and outliers were removed from the genome-wide association studies.

### 3.9. Markers Associated with Biomass Yield and Plant Height

Using combined data from the two growing seasons, eight SilicoDArt markers were found to be associated with biomass yield. Of these markers, two were detected by three of the models (FarmCPU, BLINK and GLM), whereas one was detected by both Blink and GLM models (Figure 6 and Table 3). In 2015, one SilicoDArT marker on Chr1 and one on Chr8 was associated with biomass yield and plant height, respectively, using the BLINK model (Supplementary Figure S2A, Supplementary Table S6), whereas no SNP marker was found to be associated with biomass yield or plant height. In 2016, six silicoDArT markers were found to be associated with biomass yield (Supplementary Figure S2B, Supplementary Table S6). Of these markers, one marker on Chr8 was detected by all three of the models (FarmCPU, BLINK and GLM).

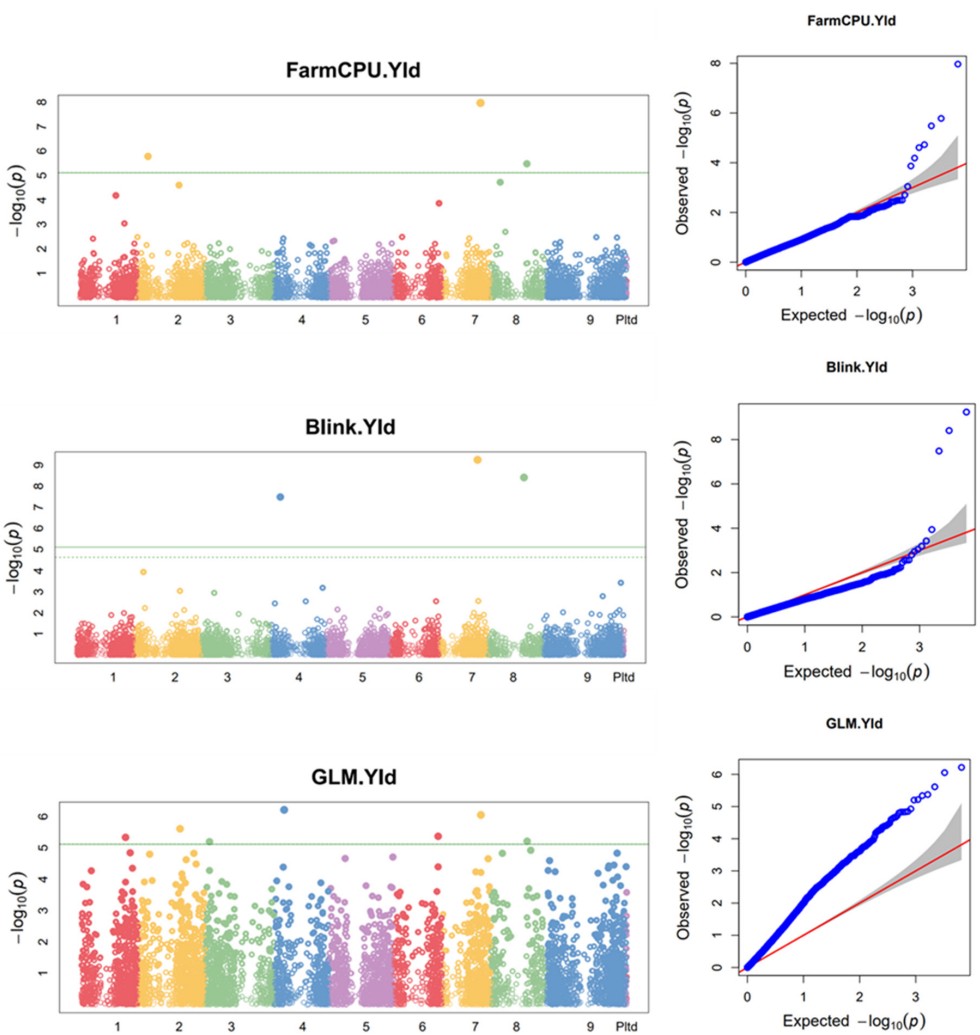

**Figure 6.** SilicoDArT markers associated with biomass yield using the combined data from the 2015 and 2016 growing seasons. On the Manhattan plots, the *x*-axis is the code of the chromosomes, and the *y*-axis is the negative log base 10 of the *p*-values. The green horizontal line indicates the significance level. QQ plot: the *y*-axis is the observed negative base 10 logarithm of the *p*-values, and the *x*-axis is the expected observed negative base 10 logarithm of the *p*-values.

**Table 3.** List of SilicoDArT markers associated with biomass yield using the combined data from the 2015 and 2016 growing seasons.

| No. | Model | Marker ID | Marker Sequence | RefSeq Sequence | Chr | pos | Allele | Minor Allele | maf | *p*-Value | R² without SNP | R² with SNP | R²* | FDR Adjusted *p* Values | Effect | Remark |
|---|---|---|---|---|---|---|---|---|---|---|---|---|---|---|---|---|
| 1 | GLM | 30838261 | TGCAGGT TTGAGGCT TGTCAGTGT GCTCGTCC CCTTGTGC CGACCTTT CCCAGGCG TC CCTGTC CGAGA | NC_028450.1 | 1 | 32,744,739 | 0/1 | 1 | 0.079 | $4.60 \times 10^{-6}$ | 0.296 | 0.445 | 0.149 | 0.006 | 2009.364 | Minor allele has positive effect |
| 2 | FarmCPU | 30921428 | TGCAGCA AATACTTA CCAGAGCA CAGGTTGC CAGAAAA TATTGTT GCAACAA CAAGTGCT GCTGATGCT | NC_028451.1 | 2 | 8,049,803 | 0/1 | 1 | 0.097 | $1.65 \times 10^{-6}$ | NA | NA | NA | 0.005 | −969.259 | Minor allele has negative effect |
| 3 | GLM | 30912865 | TGCAGAGA GTTGCAAA ACGTATCGA AACAAATGT TGGAGACTT GCCGTGGG GTGAGGTG AAGACG GACT | NC_028451.1 | 2 | 30,749,442 | 0/1 | 1 | 0.097 | $2.45 \times 10^{-6}$ | 0.296 | 0.455 | 0.158 | 0.005 | 1608.405 | Minor allele has positive effect |
| 4 | GLM | 30829864 | TGCAGGCC GATCACGC TGTACGCC ATGTGACC CAGCCGC GACGCCAC CTGCACCGC GAACCGC AAAATG | NC_028452.1 | 3 | 3,213,526 | 0/1 | 1 | 0.118 | $6.32 \times 10^{-6}$ | 0.296 | 0.441 | 0.144 | 0.006 | −1985.889 | Minor allele has negative effect |
| 5 | GLM | 30944290 | TGCAGCTG CTCCACTG TTTTCGCAC TGCTGAAC TGTTCTTCT CTAACTGA AGAATATTTG TGGGCAACC | NC_028453.1 | 4 | 7,437,264 | 0/1 | 1 | 0.075 | $6.10 \times 10^{-7}$ | 0.296 | 0.476 | 0.180 | 0.003 | 1779.373 | Minor allele has positive effect |

**Table 3.** *Cont.*

| No. | Model | Marker ID | Marker Sequence | RefSeq Sequence | Chr | pos | Allele | Minor Allele | maf | *p*-Value | R² without SNP | R² with SNP | R²* | FDR Adjusted *p* Values | Effect | Remark |
|---|---|---|---|---|---|---|---|---|---|---|---|---|---|---|---|---|
| 6 | Blink | 30944290 | TGCAGCTG CTCCACTGT TTTCGCACT GCTGAACT GTTCTTCTC TAACTGAAG AATATTTGTG GGCAACC | NC_028453.1 | 4 | 7,437,264 | 0/1 | 1 | 0.075 | $3.27 \times 10^{-8}$ | NA | NA | NA | 0.000 | NA | |
| 7 | GLM | 30846885 | TGCAGAG AGAGGGA GAGAGAG GCTATCCT ACTATGCA ACGGTCAA AAGGCTTC AAAGGAGG AGAAATCA | NC_028455.1 | 6 | 33,041,360 | 0/1 | 1 | 0.105 | $4.25 \times 10^{-6}$ | 0.296 | 0.447 | 0.150 | 0.006 | −1877.678 | Minor allele has negative effect |
| 8 | GLM | 30838332 | TGCAGTCC TAAACACCA GCACAGCA CTCTCCTCT CCTTCCATC CCTAACATA CATCATCA GCGATACAG | NC_028456.1 | 7 | 28,411,310 | 0/1 | 1 | 0.079 | $8.87 \times 10^{-7}$ | 0.296 | 0.470 | 0.174 | 0.003 | 1764.365 | Minor allele has positive effect |
| 9 | FarmCPU | 30838332 | TGCAGTCCT AAACACC AGCACAGCA CTCTCCTCT CCTTCCATC CCTAACATA CATCATCAG CGATACAG | NC_028456.1 | 7 | 28,411,310 | 0/1 | 1 | 0.079 | $1.08 \times 10^{-8}$ | NA | NA | NA | 0.000 | 1528.874 | Minor allele has positive effect |
| 10 | Blink | 30838332 | TGCAGTCCT AAACACCA GCACAGCAC TCTCCTCTC CTTCCATCC CTAACATAC ATCATCAG CGATACAG | NC_028456.1 | 7 | 28,411,310 | 0/1 | 1 | 0.079 | $5.72 \times 10^{-10}$ | NA | NA | NA | 0.000 | NA | |

**Table 3.** *Cont.*

| No. | Model | Marker ID | Marker Sequence | RefSeq Sequence | Chr | pos | Allele | Minor Allele | maf | *p*-Value | $R^2$ without SNP | $R^2$ with SNP | $R^{2*}$ | FDR Adjusted *p* Values | Effect | Remark |
|---|---|---|---|---|---|---|---|---|---|---|---|---|---|---|---|---|
| 11 | GLM | 30846154 | TGCAGTCT CCCAATCT CCCGTGGG AGCTCTGT GATTTGATC GCAGTCCT TGAGATCCA GATACC TAAGC | NC_028457.1 | 8 | 26,442,566 | 0/1 | 1 | 0.088 | $6.10 \times 10^{-6}$ | 0.296 | 0.441 | 0.145 | 0.006 | −1463.693 | Minor allele has negative effect |
| 12 | FarmCPU | 30846154 | TGCAGTCT CCCAATCT CCCGTGG GAGCTCT GTGATTTG ATCGCAGT CCTTGAGAT CCAGATA CCTAAGC | NC_028457.1 | 8 | 26,442,566 | 0/1 | 1 | 0.088 | $3.30 \times 10^{-6}$ | NA | NA | NA | 0.007 | −853.891 | Minor allele has negative effect |
| 13 | Blink | 30846154 | TGCAGTCT CCCAATCT CCCGTGGG AGCTCTGT GATTTGAT CGCAGTCC TTGAGATC CAGATAC CTAAGC | NC_028457.1 | 8 | 26,442,566 | 0/1 | 1 | 0.088 | $3.91 \times 10^{-9}$ | NA | NA | NA | 0.000 | NA | |

\* $R^2 = R^2$ with SNP $- R^2$ without SNP.

Using the combined data, three SNP markers were found to be associated with biomass yield (Figure 7 and Table 4). Two of these markers were detected with the GLM model, whereas the other one was detected using the FarmCPU model. Using 2016 data, three SNP markers were associated with biomass yield (Supplementary Figure S3, Supplementary Table S7), of which one, on Chr1, was detected using both BLINK and GLM models. The other two SNP markers associated with biomass yield were located on Chr3.

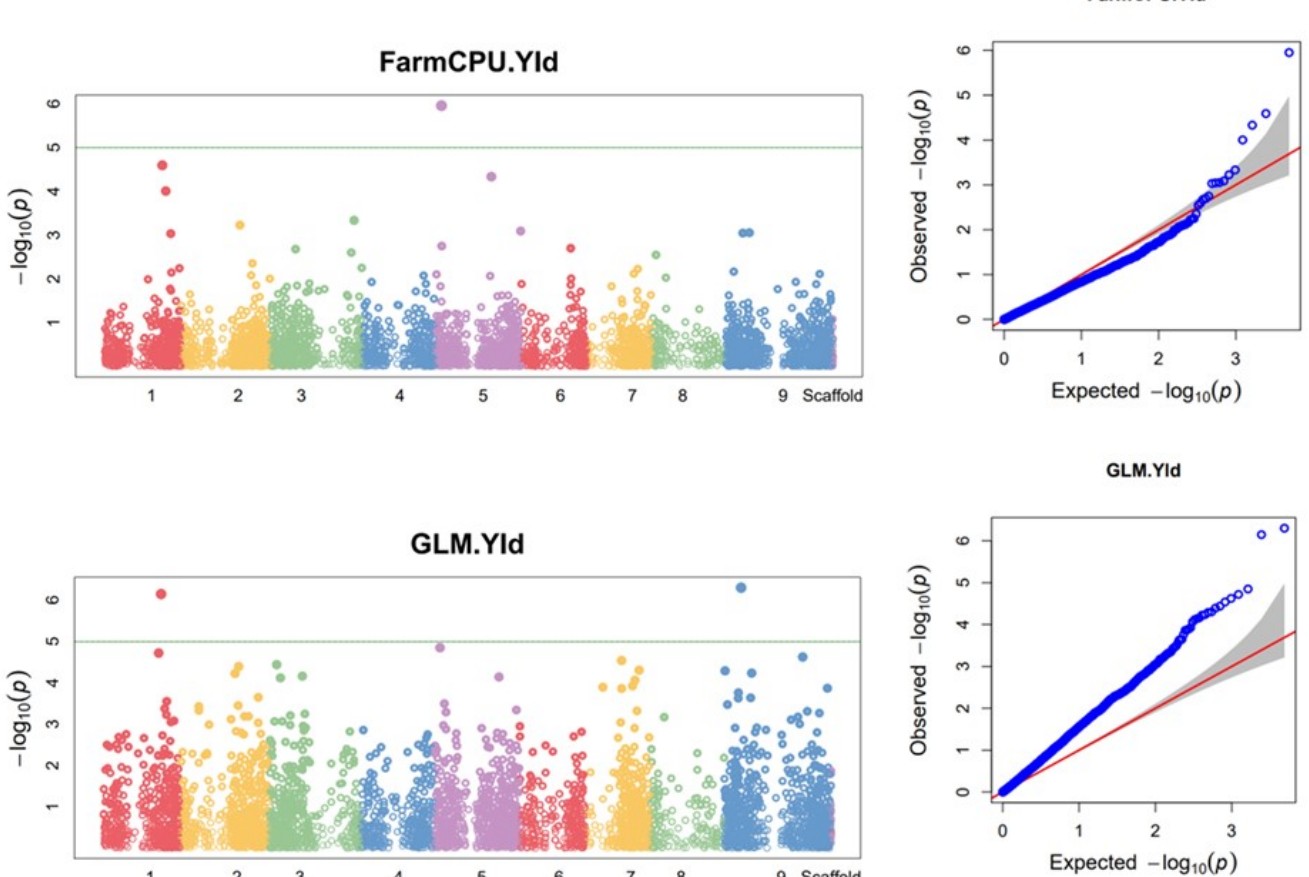

**Figure 7.** SNP markers associated with biomass yield using the combined data from the 2015 and 2016 growing seasons. On the Manhattan plots, the *x*-axis is the code of the chromosomes, and the *y*-axis is the negative log base 10 of the *p*-values. The green horizontal line indicates the significance level. QQ plot: the *y*-axis is the observed negative base 10 logarithm of the *p*-values, and the *x*-axis is the expected observed negative base 10 logarithm of the *p*-values.

### 3.10. Markers Associated with Feed Quality Traits

Using the combined data from the two growing seasons, four SilicoDArT markers were found to be associated with feed quality traits. Of these markers, two were associated with both ADF and TDN (Figure 8 and Table 5). In addition, some additional markers were also found to be associated with feed quality traits using individual season data. In the 2015 season, four SilicoDArT markers were associated with CP using the BLINK model (Supplementary Figure S4A, Supplementary Table S8), whereas no other SNP marker was found to be associated with any of the feed quality traits in 2015. In 2016, two and three SilicoDArT markers were associated with CP and TDN, respectively (Supplementary Figure S4B, Supplementary Table S8).

**Table 4.** List of SNP markers associated with biomass yield using the combined data from the 2015 and 2016 growing seasons.

| No. | Model | Marker ID | Marker Sequence | RefSeq Sequence | Chr | pos | Alleles | Minor Allele | maf | *p*-Value | R² without SNP | R² with SNP | R²* | FDR Adjusted *p* Values | Effect | Remark |
|---|---|---|---|---|---|---|---|---|---|---|---|---|---|---|---|---|
| 1 | GLM | 30964292-59-G/A | TGCAGCTC AGAGCAGT ACGACGCC ATGGCGAT CTCGGCGC CCTTGAACC CGTAGTCCA GGCTCGG GTTG | NC_028450.1 | 1 | 31,786,540 | G/A | A | 0.179 | $7.11 \times 10^{-07}$ | 0.437 | 0.571 | 0.314 | 0.002 | −1149.039 | Minor allele has a negative effect |
| 2 | FarmCPU | 30935961-51-C/T | TGCAGATC TACTAAAAT CTAGCCGC GCCAGCAG CGACGCGA ACCGCTAA ATCCACCC AAACCT AGCACC | NC_028454.1 | 5 | 3,450,894 | C/T | T | 0.058 | $1.12 \times 10^{-06}$ | NA | NA | NA | 0.006 | 992.590 | Minor allele has a positive effect |
| 3 | GLM | 30882610-38-G/A | TGCAGCG TGCGGCAG CAGACCAG ATCCGTCG GGTTGAA GTTCACCG | NC_028458.1 | 9 | 9,428,069 | G/A | A | 0.079 | $4.99 \times 10^{-07}$ | 0.437 | 0.575 | 0.138 | 0.002 | 1629.385 | Minor allele has a positive effect |

* $R^2 = R^2$ with SNP $- R^2$ without SNP.

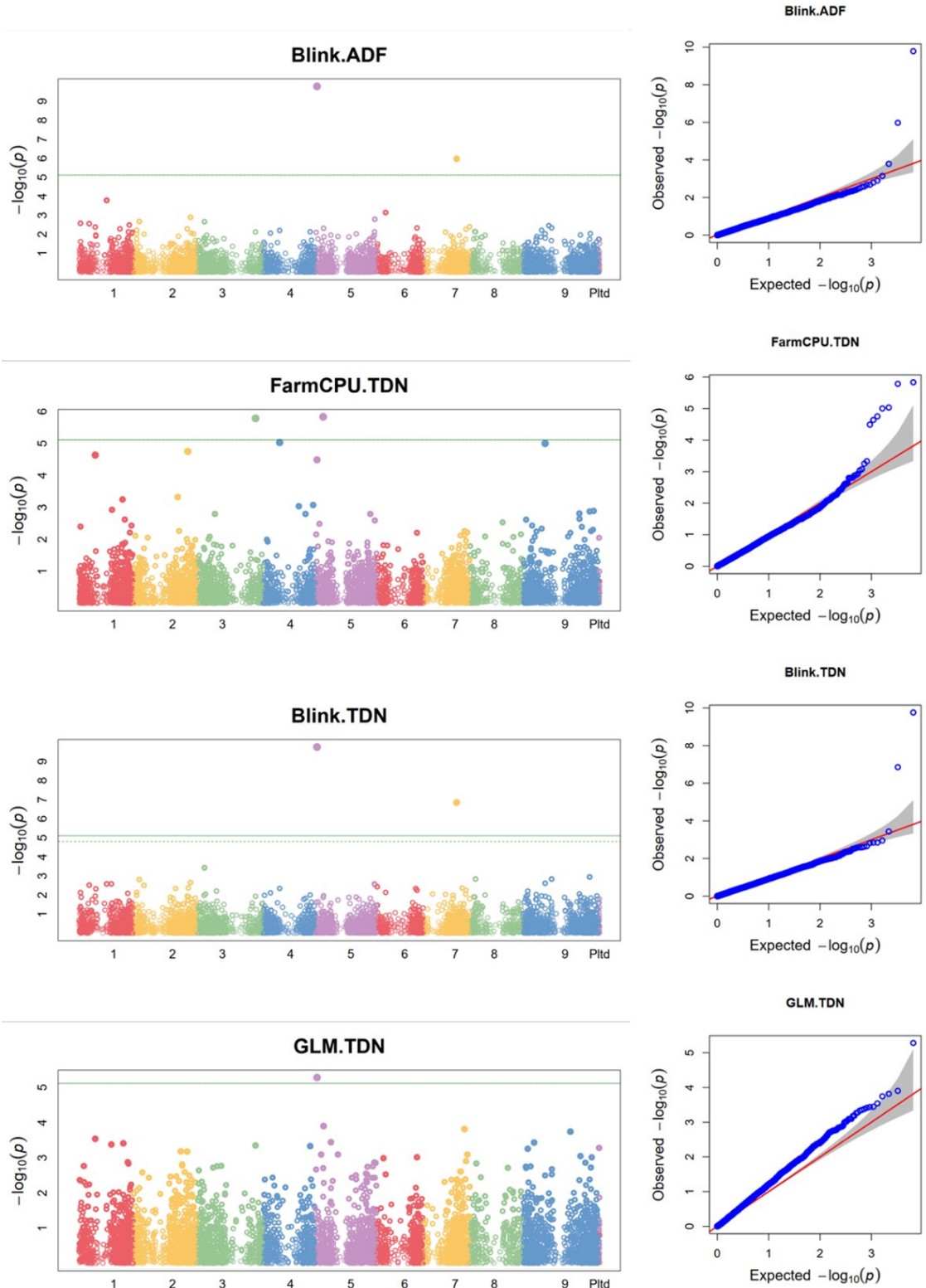

**Figure 8.** SilicoDArT markers associated with feed quality traits using the combined data from the 2015 and 2016 growing seasons. On the Manhattan plots, the *x*-axis is the code of the chromosomes, and the *y*-axis is the negative log base 10 of the *p*-values. The green horizontal line indicates the significance level. QQ plot: the *y*-axis is the observed negative base 10 logarithm of the *p*-values, and the *x*-axis is the expected observed negative base 10 logarithm of the *p*-values.

**Table 5.** List of SilicoDArT markers associated with feed quality traits using combined data from the 2015 and 2015 growing seasons.

| No. | Trait | Model | Marker ID | Marker Sequence | RefSeq Sequence | Chr | pos | Allele | Minor Allele | maf | *p*-Value | R² without SNP | R² with SNP | R²* | FDR Adjusted *p* Values | Effect | Remark |
|---|---|---|---|---|---|---|---|---|---|---|---|---|---|---|---|---|---|
| 1 | TDN | FarmCPU | 30930072 | TGCAGC TGGCGTC GGCGACG GCGTGCG TCGCGCT GTCGGCG GCGCGGC TCGCCCG | NC_028452.1 | 3 | 44,894,692 | 0/1 | 0 | 0.32 | $1.64 \times 10^{-6}$ | NA | NA | NA | 0.005 | −0.818 | Minor allele has negative effect |
| | ADF | Blink | 30879386 | TGCAGT AGTGGC GGTGGA CTACGAC GCCTCCC CCTGCGA GCACATC ATATCCC AGACGC CTGCTCG ACG | NC_028454.1 | 5 | 1,406,759 | 0/1 | 0 | 0.272 | $1.64 \times 10^{-10}$ | NA | NA | NA | 0.000 | NA | |
| 2 | TDN | GLM | 30879386 | TGCAGT AGTGGC GGTGGA CTACGA CGCCTCC CCCTGCG AGCACAT CATATCC CAGACGC CTGCTC GACG | NC_028454.1 | 5 | 1,406,759 | 0/1 | 1 | 0.272 | $5.22 \times 10^{-6}$ | 0.202 | 0.368 | 0.167 | 0.034 | 1.464 | Minor allele has positive effect |
| | TDN | Blink | 30879386 | TGCAGT AGTGGCG GTGGACT ACGACGC CTCCCCC TGCGAGC ACATCAT ATCCCAG ACGCCTG CTCGACG | NC_028454.1 | 5 | 1,406,759 | 0/1 | 1 | 0.272 | 0.000 | NA | NA | NA | 0.0000 | NA | |

**Table 5.** *Cont.*

| No. | Trait | Model | Marker ID | Marker Sequence | RefSeq Sequence | Chr | pos | Allele | Minor Allele | maf | *p*-Value | R² without SNP | R² with SNP | R²* | FDR Adjusted *p* Values | Effect | Remark |
|---|---|---|---|---|---|---|---|---|---|---|---|---|---|---|---|---|---|
|  | ADF | GLM | 30879386 | TGCAGT AGTGGCG GTGGACT ACGACGC CTCCCCC TGCGAGC ACATCAT ATCCCAG ACGCCTG CTCGACG | NC_028454.1 | 5 | 1,406,759 | 0/1 | 1 | 0.272 | $3.65 \times 10^{-6}$ | 0.169 | 0.349 | 0.180 | 0.024 | −1.509 | Minor allele has negative effect |
| 3 | TDN | FarmCPU | 30841580 | TGCAGAA CGTTCAGA CTTCAAAC CACATGCT GCCGTGC GCATCAGC ACATGTG CTTGACT TGTGACCTG | NC_028454.1 | 5 | 6,158,000 | 0/1 | 1 | 0.145 | $1.47 \times 10^{-6}$ | NA | NA | NA | 0.005 | −1.130 | Minor allele has negative effect |
| 4 | ADF | Blink | 30930612 | TGCAGCTC CCGCCGTG GCAGCAC TCCAGCG CGTCCC AGCCG | NC_028456.1 | 7 | 25,606,103 | 0/1 | 1 | 0.18 | $1.06 \times 10^{-6}$ | NA | NA | NA | 0.003 | NA | NA |
|  | TDN | Blink | 30930612 | TGCAGCTC CCGCCGTG GCAGCAC TCCAGCG CGTCCCA GCCG | NC_028456.1 | 7 | 25,606,103 | 0/1 | 1 | 0.18 | $1.40 \times 10^{-7}$ | NA | NA | NA | 0.001 | NA | NA |

* R² = R² with SNP − R² without SNP.

Using the combined data, 19 SNP markers were found to be associated with CP, 11 with NDF, 6 with ADF, 19 with TDN and 7 with DMI (Figure 9 and Supplementary Table S9). In addition, using the 2016 season data, four SNP markers were found to be associated with TDN, two with CP and seven with ADF (Supplementary Figure S5 and Supplementary Table S10). One of the SNP markers associated with CP was detected using both BLINK and GLM models, and one marker associated with TDN was detected using both FarmCPU and BLINK models.

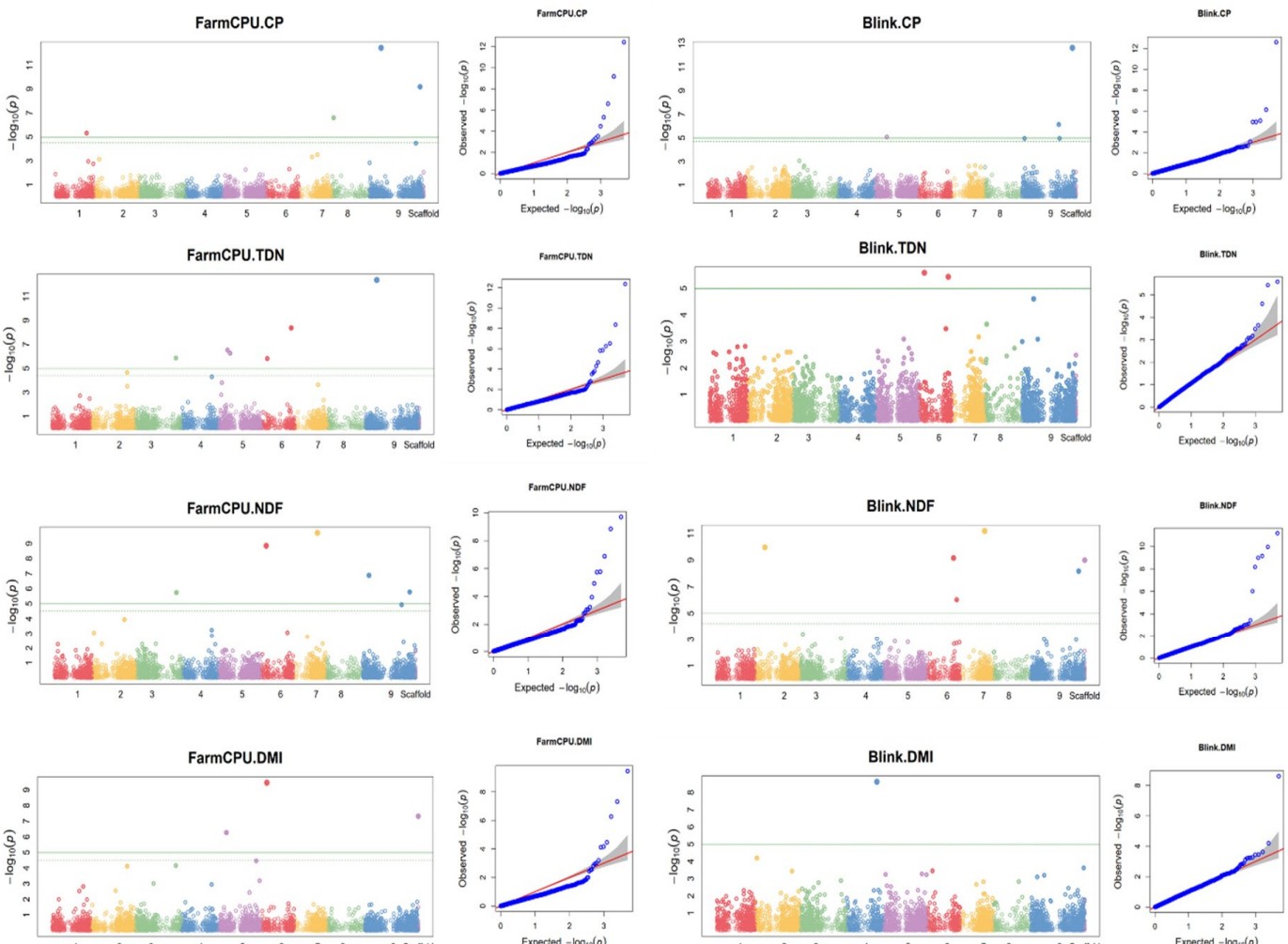

**Figure 9.** SNP markers associated with feed quality traits using the combined data from the 2015 and 2016 growing seasons. On the Manhattan plots, the *x*-axis is the code of the chromosomes, and the *y*-axis is the negative log base 10 of the *p*-values. The green horizontal line indicates the significance level. QQ plot: the *y*-axis is the observed negative base 10 logarithm of the *p*-values, and the *x*-axis is the expected observed negative base 10 logarithm of the *p*-values.

## 4. Discussion

### 4.1. Markers Associated with Feed Quality Traits

Buffel grass is an important forage grass in the tropical and subtropical regions of the world [21,22]. Substantial variation in agronomic and nutritional quality traits was observed in the buffel grass accessions, which shows the rich genetic variation embedded in the collection from which to select lines with superior performance. Year difference was also significant for all traits, indicating that a multiyear evaluation of buffel grass is essential to determine the consistent performance of the genotypes. However, the maximum biomass yield recorded in the current report is less than the biomass yield reported elsewhere [22,25,30]. The relatively lower biomass yield observed at Bishoftu

could be related to the environmental conditions and difference in management practices. On the other hand, the range of CP content of the studied accessions was wider than what has been reported elsewhere for the species [25,48]. Based on genome-wide DArTSeq markers, the collection was clustered into eight clusters [9]. The accessions in clusters I, II and III showed a low biomass yield but a relatively higher feed quality (CP, TDN and DMI) than the rest of the clusters. Cluster IV had the highest biomass yield and the tallest plants compared to the other clusters. Similarly, Jorge and colleagues [32] also studied 68 accessions and classified them based on the robustness of the plant, flowering characters and growth forms. Accordingly, some of the accessions with the highest biomass yield and tallest plants belong to the most robust and leafiest cluster group, whereas accessions with the lowest biomass yield belong to the cluster with short leaves and thin stems.

This study also revealed different levels of variability among traits. The highest PCV and GCV values were recorded for biomass yield and plant height, indicating the presence of high genetic variability for the traits. The PCV value for biomass yield was equivalent to the corresponding GCV value, whereas the PCV value for plant height was close to the GCV value. This shows the substantial contribution of genetic factors to the observed performance for both traits. Thus, directional selection might be effective to improve these two traits. On the other hand, NDF, TDN, CP, ADF and DMI showed low PCV and GCV estimates, indicating low genetic variability. PCV values for feed quality traits were greater than the corresponding GCV values, indicating the significant effect of environmental factors on the expression of these traits. In general, the evaluated accessions showed significant variation in performance. Hence, given the observed genetic and phenotypic performance variation in the collection [9,32,49], there is a potential improvement opportunity in the buffel grass germplasm to develop high-yielding climate-resilient varieties.

### 4.2. Correlation of Biomass Yield, Plant Height and Feed Quality Traits in Buffel Grass

Biomass yield and feed quality traits are important parameters in forage improvement. Understanding the relationship between biomass yield and feed quality traits and the genetic basis of their relationship would be of great importance to breeding programs. A positive correlation was observed between biomass yield and plant height (0.64 **). Biomass yield had a positive correlation with NDF (0.48 ***) and ADF (0.60 **) and a negative correlation with CP (−0.51 ***), TDN (−0.59 ***) and DMI (−0.48 ***). Plant height also had a similar trend in correlation with feed quality traits. It is also worth noting that DMI and TDN had a strong negative correlation with NDF (−0.999 ***, and −0.870 ***, respectively). NDF also had a similar correlation with DMI (−0.981 ***) and TDN (−0.885 ***). The observed relationship between the traits was very similar during the two growing seasons. The correlation observed between biomass yield and plant height, as well as biomass yield and feed quality traits, have implications for improvement programs. For example, plant height could be used a good indicator for biomass yield under field conditions. However, the negative correlation between biomass yield and feed quality traits (CP, TDN and DMI) needs special attention in improvement programs, as varieties with a higher biomass yield might be poor in feed quality. Thus, the high-biomass-yielding accessions and accessions that produce high CP content would be the candidate accessions for further field performance evaluation in different tropical agroecologies and seasons.

### 4.3. Marker–Trait Associations in Buffel Grass

In this study, we used a buffel grass germplasm collection in the forage genebank at ILRI and conducted GWAS to identify marker–trait associations. Several marker–trait associations were identified. Using the combined data from the two seasons, three SNP and eight SilicoDArT markers were found to be associated with biomass yield. A total of nine markers (six SilicoDArT and three SNP) were found to be associated with biomass yield in the 2016 growing season. One of the SNP and two of the SilicoDArT markers were detected both using the 2016 season and the combined data. In the 2015 growing season, one SilicoDArT marker was associated with biomass yield. One SilicoDArT marker was

associated with plant height in 2015. No marker was associated with plant height using the 2016 season and the combined data.

Using the combined data, four SilicoDArT markers were found to be associated with ADF and TDN, whereas no marker was found to be associated with the other feed quality traits (CP, NDF and DMI). One of the four markers was also detected using individual season data, and it was associated with CP in 2015 and TDN in 2016. A total of eight SilicoDArT markers (four in 2015 associated with CP, two in 2016 associated with CP and three in 2016 associated with TDN) were found to be associated with feed quality traits.

Using the combined data, a total of 42 SNP markers were associated with feed quality traits, of which four were also detected using the 2016 season data. Of the four markers detected using both combined and 2016 season data, two were associated with the same trait (one with ADF and the other with TDN). The other two markers were associated with different traits depending on the dataset. Seven SNP markers were associated with DMI using the combined data, whereas no marker was found to be associated with the trait using the individual season data. Thirteen SNP markers were found to be associated with feed quality, using data from the 2016 growing season. Of these markers, four were associated with TDN, two with CP and seven with ADF. The different marker–trait associations identified between the two growing seasons (2015 and 2016) could be related to the difference in weather conditions (Supplementary Table S11). For example, the average monthly rainfall of the location during the months of July to September was 117 mm in 2015 and 142 mm in 2016. In addition, in 2015, the minimum and maximum daily temperature during July to August was 12 °C and 26 °C whereas it was 13 °C and 28 °C, respectively, during the same months in 2016. The variation in growing conditions would affect the performance of the genotypes and result in variation in the marker–trait associations for the different years. Another reason could be that the plants were well established during the second season and therefore more able to reach the crops' genetic potential in terms of performance.

*4.4. Genome-Wide Distribution and Co-Localisation of the Marker–Trait Associations*

Except for a few studies with conventional molecular markers [50], genomic studies are limited in buffel grass. A reference genome has not been developed to date. The lack of its own reference genome has hindered the mapping and selection of genome-wide representative markers for further molecular studies. As a result, the reference genome of *Setaria italica* [36] was used to map the generated markers. However, only a small percentage of the total markers was successfully mapped [9]. Despite this challenge, we conducted a GWAS using the mapped markers and identified several marker–trait associations with $R^2$ values ranging from 0.138 to 0.236. The identified marker–trait associations were distributed across the different chromosomes of the *Setaria italica* genome (Figure 10). On Chr1, three SilicoDArT markers (one for CP and two for biomass yield) and four SNPs (one associated with CP and TDN and one each for biomass yield, CP and ADF) were identified. The SNP associated with ADF was detected using the 2016 season whereas the SNPs with biomass yield, CP and TDN were identified using the combined data. Three SilicoDArT markers (one for CP and three for biomass yield using the 2016 season data) and two SNP markers (one each for CP and NDF using combined data) were identified on Chr2. Five SNP markers (two associated with biomass yield, one with TDN, one with both NDF and TDN and one with both CP and TDN) and two SilicoDArT markers (one each for yield and TDN) were located on Chr3. On Chr4, one SilicoDArT marker associated with biomass yield, and one SNP marker associated with multiple traits (ADF, TDN and DMI) were identified using the combined data. No marker on this chromosome was found to be associated with these traits using individual season data.

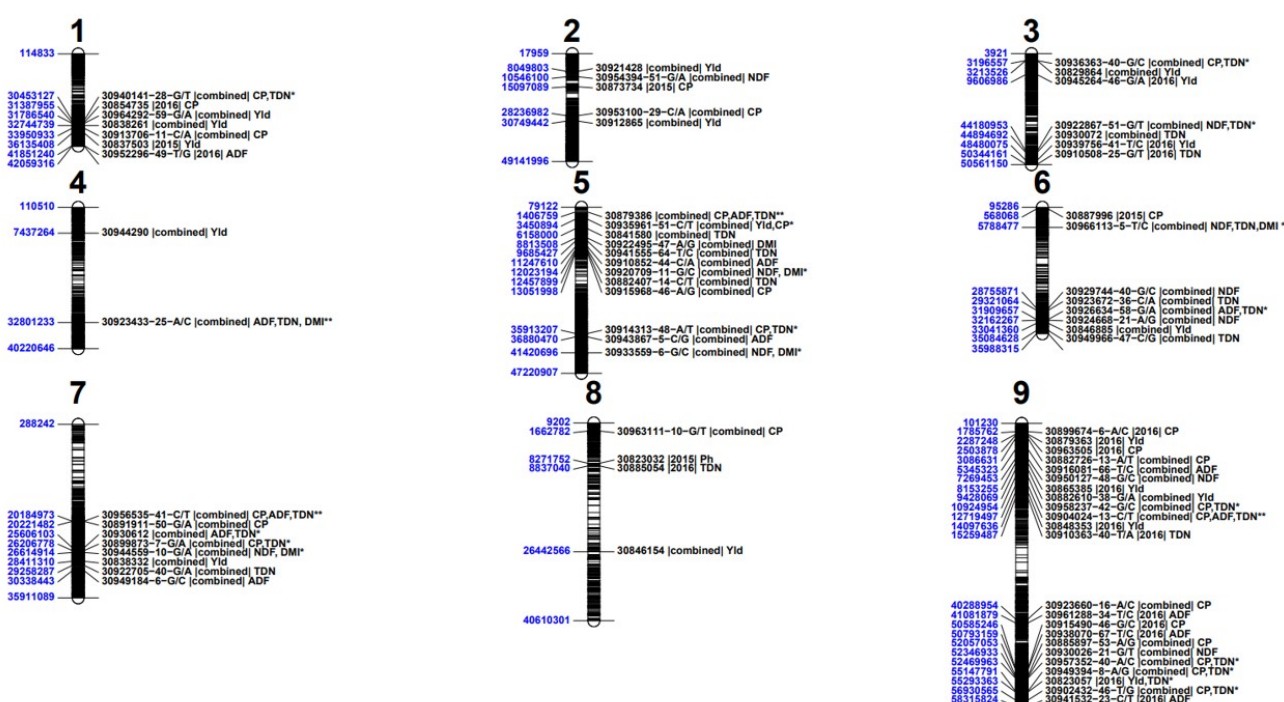

**Figure 10.** Genomic map position of the marker–trait associations. The labels on the right of the linkage groups indicate the marker code, followed by the growing season of the data used for GWAS (2015, 2016 or combined) and the traits (Yld = Biomass Yield, Ph = Plant height, CP = Crude Protein, NDF = Neutral Detergent Fibre, ADF = Acid Detergent Fibre, TDN = Total Digestible Nutrients, DMI = Dry Matter Intake). The number on the left of the linkage groups indicates the physical genomic position of the markers in base pairs. * Indicates markers associated with two traits, and ** indicates markers associated with three traits.

On Chr5, 13 markers were associated with different traits. One SilicoDArT marker was associated with both CP (2015 season data) and TDN (2016 season data, with a positive effect) whereas it was associated with ADF and TDN using the combined data. The marker showed a negative effect on ADF, whereas it had a positive effect on TDN. Another two SilicoDArT markers associated with CP (2016 season) and TDN (combined data) were also located on this chromosome. In addition to the SilicoDArT markers, using the combined data, ten SNP markers associated with different traits (one with biomass yield, three with TDN, three with DMI, three with CP, two with ADF and two with NDF) were also found on this chromosome. Three of these SNP markers were associated with two different traits, of which two were associated with NDF and DMI, with negative and positive effects on the traits, respectively, and the other one with CP and TDN, with positive effects on both traits.

On Chr6, there were two SilicoDArT markers (one each associated with CP and biomass yield) and six SNP markers (two with TDN, two with ADF, one with both TDN and ADF and one with TDN and DMI) associated with different traits. One of the SNP markers was associated with three feed quality traits (NDF, TDN and DMI) whereas one was associated with both ADF and TDN. The marker associated with NDF, TDN and DMI had a negative effect on NDF, whereas it had a positive effect on TDN and DMI. Six SNP and two SilicoDArT markers associated with different traits were located on Chr7. One of these markers was associated with three feed quality traits (CP, ADF and TDN), whereas the other three markers were associated with two different traits. The SNP marker associated with NDF and DMI had negative and positive effects, respectively, whereas the SNP marker associated with CP and TDN had a positive effect on both traits. Three SilicoDArT markers (one each for plant height, biomass yield, and TDN) and one SNP marker associated with CP were located on Chr8. A total of 23 markers associated with traits (18 SNP and 5 SilicoDArT) were located on Chr9. Of these markers, nine were

associated with CP (eight markers with a negative effect), five with ADF (three with a positive effect based on the 2016 season data), two with NDF (both with a negative effect), one with biomass yield (with a positive effect based on the combined data from the two seasons) and five with TDN. Among the SilicoDArT markers, one was associated with both biomass yield and TDN, two with CP and three with biomass yield. One of the SNP markers was associated with three feed quality traits (CP, ADF and TDN), whereas four SNP markers were associated with two of the traits.

In a few cases, a single marker was associated with two and three traits, or markers associated with two different traits were closely located on the same chromosome. Markers associated with three traits were found on Chr4, Chr5, Chr6, Chr7 and Chr9. The markers on Chr4 and Chr6 were associated with ADF, TDN and DMI, whereas the markers on Chr5, Chr7 and Chr9 were associated with CP, ADF and TDN. In addition, several markers associated with two traits were also found on Chr1, Chr3, Chr5, Chr6, Chr7 and Chr9. For example, a SilicoDArT marker on Chr5 was associated with both CP and TDN, whereas another SilicoDArT marker on Chr9 was associated with both biomass yield and TDN. Closely located marker–trait associations were also found on six of the nine chromosomes. On Chr1, a SilicoDArT marker associated with CP and a SNP marker associated with biomass yield were located at 398,585 bp from each other. Similarly, on Chr8, markers associated with plant height and biomass yield had a physical distance of 565,288 bp from each other. Among the markers on Chr9, a SNP marker associated with CP and a SilicoDArT marker associated with biomass yield were located at 501,486 bp from each other. Other closely located marker–trait associations were also found on Chr1 (biomass yield and CP/TDN), Chr3 (ADF/TDN and NDF), Chr5 (TDN and NDF/DMI), Chr6 (NDF and ADF/TDN), Chr7 (CP and CP/ADF/TDN) and Chr9 (biomass yield and CP, CP and ADF, NDF and CP/TDN). In total, 78 marker–trait associations distributed across the different chromosomes (one based on both individual growing season and combined data, 47 based on combined data only, 21 based on individual growing season data only and 9 based on both combined and 2016 growing season data) were identified in this study. The largest number of marker–trait associations were located on Chr9, whereas the lowest number of markers were located on Chr4. In terms of traits, the largest number of markers was associated with TDN followed by CP and biomass yield. The generated information on the genome distribution of the marker–trait associations will be a useful resource for future improvement programs in this important tropical forage. Furthermore, an additional study is required to validate the associations and co-localisation of the identified markers. In line with this suggestion, it is very important to develop a buffel grass reference genome to facilitate genomic studies and the development of markers for efficient marker-assisted selection/breeding. The lack of a reference genome is one of the main challenges to genomic studies of tropical forages such as buffel grass. In this study, we used the reference genome of *Setaria italica*, a model grass species, to map the generated buffel grass DArTSeq markers, which enabled us to map only a small percentage of the generated markers. On several occasions, developing and using a species-specific reference genome increased the efficiency of mappable markers and the discovery of marker–trait associations. Thus, developing a species-specific reference genome will increase the number of mappable markers and thereby improve the discovery and accuracy of the marker–trait associations in this drought-tolerant tropical forage.

### 4.5. Marker–Trait Association in Functional Putative Genomic Regions

Some of the identified marker–trait associations were in genomic regions related to key enzymes and proteins involved in different biochemical reactions and processes in plants. Among the identified SNP markers associated with biomass yield, one was located on Chr1 in the genomic region linked to a gene encoding a Phenylalanine ammonia-lyase (PAL)-like protein. PAL catalyses the deamination of phenylalanine to produce trans-cinnamic acid, a precursor of lignins, flavonoids and coumarins, and it is a key enzyme that induces the synthesis of salicylic acid, which causes systemic resistance in many plants [51,52]. A recent

study has shown that PAL-knockdown plants in the model grass *Brachypodium distachyon* exhibited delayed development and reduced root growth, as well as increased susceptibility to diseases [53]. Another marker associated with biomass yield was located on Chr3 in the region related to a gene encoding a U-box domain containing protein 1. This protein is in the family of ubiquitin ligase (E3) enzymes that are involved in various biological processes and in the stress response in plants [54]. Similarly, the SilicoDArT marker associated with plant height was located on Chr8 in the genomic region harbouring a gene annotated as a *Setaria italica* ankyrin-1 protein. This protein family is conserved in plants and involved in biochemical processes in response to biotic and abiotic stresses [55–57].

Several markers were found to be associated with feed quality traits. These markers were distributed over the different chromosomes of the *Setaria italica* genome. Some of the marker–trait associations were located in the genomic regions linked to different biophysiological processes in plants. One of the marker–trait (CP) associations on Chr2 was close to a gene encoding an E3 ubiquitin–protein ligase RGLG1, like in *Setaria viridis*. E3 ubiquitin–protein ligase is a family of proteins that catalyses the ubiquitination of protein substrates for targeted degradation [58], acts as central regulators in plant hormone signalling pathways [59] and has been known as an important regulator of drought stress response in plants [60]. A SilicoDArT marker associated with TDN (on Chr8) was close to a gene encoding a predicted *Setaria italica* chlorophyll a/b-binding protein CP26, chloroplast. This protein is conserved in plants and green algae and plays a key role in absorbing and transferring sunlight energy into chemical energy [61]. Both E3 ubiquitin–protein ligases and chlorophyll a/b-binding proteins are involved in many other biophysiological processes that contribute to plant growth and development.

Among the SNP markers associated with feed quality traits, the marker associated with TDN (on Chr6) was close to genes encoding a tryptophan decarboxylase and aromatic-L-amino-acid decarboxylase in grass, which are involved in many biochemical reactions, contributing to the formation of many metabolites involved in biotic and abiotic stress defence in plants [62,63]. A marker associated with CP (on Chr9) was located in the genomic region containing a gene encoding a pentatricopeptide-repeat (PPR)-containing protein. PPR proteins are one of the largest nuclear-encoded protein families in higher plants and interact with RNA to affect gene expressions necessary for organelle development [64,65]. On Chr9, another SNP marker associated with ADF was found in the genomic region harbouring a gene encoding a detoxification-40-like protein, which is believed to play a role in response to stresses in plants (e.g., the detoxification of a heavy metal, Cd(2+), in rice) [66]. In general, marker–trait associations in genomic regions containing genes linked to important enzymes and proteins were identified. This result could be used as a starting point for further study to elucidate genomic regions with genes controlling important traits such as drought tolerance, disease resistance and feed quality traits.

## 5. Conclusions and Recommendations

Here we reported the first genome-wide association study in buffel grass, an important drought-tolerant tropical forage grass. Several markers were found to be associated with biomass yield and feed quality traits. The largest number of markers was associated with TDN followed by CP and biomass yield. Some of the markers were associated with multiple traits: eight markers were associated with CP and TDN; two markers with ADF and TDN; two markers with CP, TDN and DMI; two markers with NDF and DMI; one marker with ADF, TDN and DMI; one marker with NDF, TDN and DMI; and one marker with biomass yield and TDN. Some of the associated markers were located in the genomic regions containing genes related to key biochemical processes that affect yield, stress responses and feed quality traits in plants. In general, the identified marker–trait associations will be a useful genomic resource for buffel grass genomic studies and will have a significant implication on future buffel grass improvement programs, and we recommend the following as future lines of research on buffel grass for accelerated improvement programs:

- Developing a reference genome that can be used for marker mapping and genome-wide association studies to identify major QTL for traits of interest with an improved association accuracy.
- Buffel grass has different ploidy levels. Hence, determining the ploidy level, coupled with the identification of sexually reproducing lines, will facilitate a breeding program for developing new improved varieties of this economically important forage species.
- Buffel grass is a drought-tolerant grass species. Being an underutilised crop, little is known about the genetic basis of its drought tolerance trait. Hence, it is important to study the genetic and physiological basis of drought tolerance and other important traits to develop a climate-resilient variety.
- The results of this study can also be used as a basis to develop a set of markers for future marker-assisted selection and breeding.

**Supplementary Materials:** The following supporting information can be downloaded at https://www.mdpi.com/article/10.3390/agriculture14020257/s1, Supplementary Figure S1. Histogram showing data distribution for the different traits. Supplementary Figure S2. SilicoDArT markers associated with biomass yield and plant height in the 2015 (A) and 2016 (B) growing seasons. Supplementary Figure S3. SNP markers associated with biomass yield in the 2016 growing season. Supplementary Figure S4. SilicoDArT markers associated with feed quality traits in buffel grass. Supplementary Figure S5. SNP markers associated with feed quality traits detected using different models in buffel grass. Supplementary Tables: Supplementary Table S1. Passport data of buffel grass accessions used in this study. Supplementary Table S2. Mean performance of the accessions for the two growing seasons (2015 and 2016). Supplementary Table S3. Minimum, mean and maximum performance of the accessions for the 2015 and 2016 growing seasons. Supplementary Table S4. Mean performance of the genetic clusters identified based on DArTSeq markers. Supplementary Table S5. Number of markers mapped per each chromosome. Supplementary Table S6. Sequence and chromosome position of SilicoDArT markers associated with biomass yield (YLD) and plant height (PH) using different models during the 2015 and 2016 growing seasons. Supplementary Table S7. Summary of SNP markers associated with biomass yield in 2016 growing season. Supplementary Table S8. SilicoDArT markers associated with feed quality using the 2015 and 2016 growing seasons' data. Supplementary Table S9. List of SNP markers associated with feed quality traits using the combined data from the 2015 and 2016 growing seasons. Supplementary Table S10. SNP markers associated with feed quality traits in the 2016 growing season. Supplementary Table S11. Monthly temperature and precipitation data from 2014 to 2016.

**Author Contributions:** Conceptualisation, C.S.J.; methodology, A.T.N.; formal analysis, A.T.N. and E.H.; investigation, A.T.N. and R.A.S.G.; resources, C.S.J.; data curation, A.T.N. and R.A.S.G.; writing—A.T.N.; writing—review and editing, A.T.N., M.S.M., R.A.S.G., E.H., A.M. and C.S.J.; visualisation, A.T.N.; supervision, M.S.M. and C.S.J.; funding acquisition, C.S.J. and A.M. All authors have read and agreed to the published version of the manuscript.

**Funding:** This research was conducted as part of the CGIAR Initiatives Genebanks and Sustainable Animal Productivity (SAPLING), which are supported by contributors to the CGIAR Trust Fund. CGIAR is a global research partnership for a food-secure future dedicated to transforming food, land and water systems in a climate crisis.

**Institutional Review Board Statement:** Not applicable.

**Informed Consent Statement:** Not applicable.

**Data Availability Statement:** All data generated in this study are freely available as international public goods.

**Acknowledgments:** The authors acknowledge the support of contributors to the CGIAR Trust Fund.

**Conflicts of Interest:** The authors declare no conflicts of interest. The funders had no role in the design of the study; in the collection, analyses or interpretation of data; in the writing of the manuscript; or in the decision to publish the results.

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
