# Peer review of "A Genome-Wide Association Study of Biomass Yield and Feed Quality in Buffel Grass (Cenchrus ciliaris L.)"

_agriculture, doi:10.3390/agriculture14020257_

Round 1
Reviewer 1 Report
Comments and Suggestions for Authors
I have the following suggestions and comments:
Line 105: Provide the dates for both years when the plants were clean-cut at 10 cm.
Line 115: Define NIRS
Figure 1 should be based on the annual means and combined mean and not the replications (i.e. raw data).
Line 181-187 Change as shown:
3.2. Correlation of phenotypic and feed quality traits.
There was a positive and significant correlation among biomass yield, plant height, NDF and ADF. Similarly, a positive and significant correlation was observed among CP, TDN and DMI. On the other hand, CP and DMI had a negative and significant correlation with biomass yield, plant height, NDF, and ADF (Supplementary Figure S1). shows correlation coefficients between yield, plant height and nutritional quality traits among the buffel grass accessions for the two growing seasons and the combined data.
Line 188: Section 3.3 of results should be placed prior to section the current section 3.2. Also, in this section, the authors ignored the significant genotype x season interaction for ADF. This needs to be addressed.
Section 3.5: the across season mean Figure S2 should be part of the manuscript and not a supplemental figure.
Line 335: Here you say there was a significant genotype x season interaction for all traits. This is the opposite of the ANOVA results presented that only show a significant interaction for ADF. Seasons were different for all traits but no the interaction. These statements need to be clarified.
Line 342: change “grass” to “species”.
Line 351: change “genotypes” to “traits”.
Author Response
Dear Reviewer,
Thank you so much for the constructive comments. We have addressed the comments to the best of our capacity as per the attached file.
Thank you again for you for your time and feedback.
With regards,
Alemayehu Teressa Negawo

Reviewer 2 Report
Comments and Suggestions for Authors
For a better presentation the following is suggested:
1. Lines 48-49. Authors refer to “our institute”; write the name of the institute, since the authors are affiliated to different institutes.
2. Lines 119, 155. Please be specific about the type of test used to check data normality.
3. Lines 133-137. Heritability in the broad sense is not very useful as a parameter to be involved in a genetic improvement program, as it provides limited and little predictive information on selection responses, contrary to what the authors claim in the Discussion section (lines 357-358), to achieve an effective selection it is necessary to know the partition of the additive variance, which is the one that participates in the selection response formulas. This parameter could even be removed from the manuscript without detriment to the main topic.
4. Tables 3 and 4. The sequence of each molecular marker associated with biomass production is provided. What is the specific version ("allele") of each of these markers favorable for biomass production?
5. Same situation for Table 5 regarding feed quality traits (CP, NFD, AFD).
6. Please expand the discussion on interaction markers x growing seasons to better explain why different markers were associated with traits of interest in 2015 and 2016.
7. The authors made valuable suggestions on how to move forward in future studies on this species; however, these suggestions should be relocated in the Discussion section.
Author Response
Dear Reviewer,
Thank you so much for the constructive comments. We have addressed the comments to the best of our capacity as per the attached file (please see the attachment).
Thank you again for you for your time and feedback.
With regards,
Alemayehu Teressa Negawo.
